# A Vaccine of SARS-CoV-2 S Protein RBD Induces Protective Immunity

**DOI:** 10.3390/ijms232213716

**Published:** 2022-11-08

**Authors:** Qiaoqiao Qu, Pengfei Hao, Wang Xu, Letian Li, Yuhang Jiang, Zhiqiang Xu, Jing Chen, Zihan Gao, Zhaoxia Pang, Ningyi Jin, Chang Li

**Affiliations:** 1State Key Laboratory for Conservation and Utilization of Subtropical Agro-Bioresources, Guangxi University, Nanning 530004, China; 2Research Unit of Key Technologies for Prevention and Control of Virus Zoonoses, Chinese Academy of Medical Sciences, Changchun Veterinary Research Institute, Chinese Academy of Agricultural Sciences, Changchun 130122, China

**Keywords:** severe acute respiratory syndrome coronavirus type 2 (SARS-CoV-2), coronavirus disease 2019 (COVID-19), the receptor binding domain (RBD), the six-helix bundle (6HB)

## Abstract

The pandemic of the novel severe acute respiratory syndrome coronavirus 2 (SARS-CoV-2) has posed great threat to the world in many aspects. There is an urgent requirement for an effective preventive vaccine. The receptor binding domain (RBD), located on the spike (S) gene, is responsible for binding to the angiotensin-converting enzyme 2 (ACE2) receptor of host cells. The RBD protein is an effective and safe antigen candidate. The six-helix bundle (6HB) “molecular clamp” is a novel thermally-stable trimerization domain derived from a human immunodeficiency virus (HIV) gp41 protein segment. We selected the baculovirus system to fuse and express the RBD protein and 6HB for imitating the natural trimeric structure of RBD, named RBD-6HB. Recombinant RBD-6HB was successfully obtained from the cell culture supernatant and purified to high homogeneity. The purity of the final protein preparation was more than 97%. The results showed that the protein was identified as a homogeneous polymer. Further studies showed that the RBD-6HB protein combined with AL/CpG adjuvant could stimulate animals to produce sustained high-level antibodies and establish an effective protective barrier to protect mice from challenges. Our findings highlight the importance of trimerized SARS-CoV-2 S protein RBD in designing SARS-CoV-2 vaccines and provide a rationale for developing a protective vaccine through the induction of antibodies against the RBD domain.

## 1. Introduction

Severe acute respiratory syndrome coronavirus 2 (SARS-CoV-2) is the causative agent of COVID-19, which escalated into a global pandemic in 2020. For the past two years, due to the COVID-19 pandemic, people have been inconvenienced to travel, and economic development has been hampered. Vaccines are still an important means of preventing COVID-19. A subunit vaccine is a recombinant protein vaccine containing specific viral antigens. With the help of vaccine adjuvants, the immune system can recognize antigenic proteins with high immunogenicity and produce specific antibodies against the immunodominant antigenic epitopes with high safety and immunogenicity. Up to now, SARS-CoV-2 subunit vaccines have been approved for marketing or emergency use worldwide, and dozens of other subunit vaccines are in clinical trials [1]. Subunit vaccines use a single protein antigen molecule to induce an immune response. Therefore, in developing subunit vaccines, the DNA encoding the specific antigen with immune activity must first be identified, and the surface glycoprotein encoding gene of the pathogen is generally selected; for easily mutated viruses, each subtype can be selected. The major protective antigen gene sequence of the shared core protein.

SARS-CoV-2 is an enveloped, positive-sense, single-stranded RNA virus of the genus *Betacoronavirus*. The coronavirus is made up of nucleocapsid (N), membrane (M), envelope (E), and spike (S) proteins, which are structural proteins. The entry steps of the viral particles are mediated by the S glycoprotein (encompassing attachment to the host cell membrane and fusion). S protein is assembled as a homotrimer and inserted in multiple copies into the virion membrane, giving it its crown-like appearance. Therefore, the S protein on the mature virion consists of two non-covalently associated subunits: the S1 subunit binds ACE2, and the S2 subunit anchors the S protein to the membrane [2,3]. The receptor binding domain (RBD), located at the spike (S) gene, is responsible for binding to the angiotensin-converting enzyme 2 (ACE2) receptor of the host cells [4,5,6]. Importantly the RBD is also the primary target of the neutralizing antibodies elicited by natural infection or vaccination. The RBD protein is an effective and safe antigen candidate [7].

In addition, studies have shown that removing glycosylation sites in the RBD protein can increase protein expression and maintain its antigenicity and functionality. RBD-sc-dimer with a tandem repeat single-chain has proved to induce effective protection against SARS-CoV-2 [8,9]. Therefore, we speculate that trimers similar to native structures may increase the immunogenicity of RBD. The six-helix bundle (6HB) “molecular clamp” is a novel thermally-stable trimerization domain derived from a human immunodeficiency virus (HIV) gp41 protein segment. In this study, 6HB was added to the C-terminus of RBD to make RBD form a trimeric structure to mimic its natural structure [10].

Second, the application of recombinant DNA technology to develop subunit vaccines requires a suitable expression system to produce gene products. The expression systems applied for subunit vaccine production mainly include Escherichia coli, Bacillus subtilis, yeast, insect cells, mammalian cells, transgenic plants, and transgenic animals. Expression products in prokaryotic systems (such as *E. coli*) usually cannot be processed and folded correctly, affecting the conformation and immunogenicity of expression. While eukaryotic systems (baculovirus, insect cells, and yeast) can overcome the above shortcomings, the expression product is better, and the application is more. It only contains one or several specific immune-related protein components that highly express pathogens in the eukaryotic expression system, cannot proliferate *in vivo*, has no risk of proliferation, and is safe and reliable. The baculovirus expression system belongs to the eukaryotic expression system, which has the advantages of safety and high efficiency, and the expressed protein is closer to the natural structure. 

Based on this, we selected the baculovirus system to express the fusion of the deglycosylated RBD protein (RBDΔN1) of SARS-CoV-2 and 6HB and combined it with adjuvant to carry out protective immune experiments in hACE2 mice.

## 2. Results

### 2.1. Recombinant Baculovirus Expressing Exogenous Genes Was Rescued in Sf9 Cells Using the Recombinant Bacmid

The rB-SARS-CoV2-RBD-6HB and empty vector recombinant bacmid (Figure 1A) were identified by RT-PCR using pUC/M13F and pUC/M13R primers. The results showed that rB-SARS-CoV2-RBD-6HB can amplify a fragment with a length of about 3518 bp, and the empty vector can amplify a fragment with a length of about 2300 bp, which is in line with expectations. (Figure 1C). After rBV-SARS-CoV2-RBD-6HB was inoculated into sf9 cells, compared with the negative control group (MOCK), a significant cytopathic effect (CPE) was observed under the microscope 72 h later (cells fell off a lot, and adherent cells were sparse and swollen) (Figure 1D). 

The sf9 cells infected with rBV-SARS-CoV2-RBD-6HB for 72 h were harvested. After total RNA extraction, reverse transcription, and PCR identification. The target band with a length of 1260 bp can be seen, which is in line with expectations (Figure 1C).

### 2.2. Protein Expression and Purification

The adherent sf9 cells infected with recombinant baculovirus were harvested post 72 h, sonicated, and subjected to SDS-PAGE electrophoresis, and the protein was transferred to NC membrane, blocked with 5% skim milk, incubated with primary antibody (RBD mouse polyclonal antibody), and goat anti-mouse. After incubation with the secondary antibody, a band with a size aligned with expectations (Figure 2A). These results indicated that both RBD-6HB, RBD, and 6HB proteins were effectively expressed in the infected Sf9 cells. 

The culture medium of sf9 suspension cells infected with rBV-SARS-CoV2-RBD-6HB for 96 h was harvested, eluted after column equilibration, protein adsorption, washing, and the purified protein was obtained and subjected to SDS-PAGE electrophoresis and Coomassie brilliant blue staining (Figure 2B). After purification, the protein purity reached 97% (Figure 2B). These results indicate that we obtained RBD-6HB protein with higher purity. The acquisition of RBD protein is the same as RBD-6HB protein. After purification, the protein purity reached 99% (Figure 2C). These results indicate that we obtained RBD-6HB protein with higher purity. We treated RBD-6HB with reducing and non-reducing buffers to determine whether RBD-6HB protein could form a polymer. The RBD-6HB protein was expressed in sf9 cells and purified as single trimer-sized proteins (molecular weight ∼120 kDa) as verified by SDS-PAGE (Figure 2B) and Western blot (Figure 2B). 

### 2.3. Specific Antibodies in hACE2 Mice

Female 6-week-old hACE2 mice (Cyagen, Suzhou, China) were randomly divided into three groups in each experiment and intramuscularly immunized with indicated protein and adjuvant. Two weeks after the first immunity, conduct the booster immunity at the same dose as the first immunity (Figure 3A). Specific antibody levels increased significantly after booster immunizations (Figure 3B). The antibody titer in the RBD-6HB group was higher than those in the RBD group, and we did not detect specific antibodies in the 6HB group. After booster immunizations showed stronger specific antibody responses in mice (Figure 3C–E), the highest dilution was 1:25,600.

The body weight of mice did not change significantly after immunization, and we also observed the food intake and activity of the mice after the vaccine injection. There is no significant difference between the vaccine and control groups, indicating that the vaccine is less irritating (Figure 3F). 

### 2.4. Complete Protection against Challenge by RBD-6HB Vaccination

Animal experiments were performed as previously described. Compared with the control 6HB group, the body weight of RBD-6HB group mice fluctuated slightly after challenge, but changed little (Figure 4A). According to the results of the N gene qPCR test, RBD-6HB with the adjuvant group was 100 times lower than those of the 6HB group (*p* < 0.0006) (Figure 4B). RBD with the adjuvant group was 50 times lower than those of the 6HB group (*p* < 0.0056) (Figure 4B). It may indicate that the candidate vaccine had the effect of neutralizing the virus.

The E gene test results showed that RBD-6HB with the adjuvant group was 200 times lower than the 6HB group (*p* < 0.0001) (Figure 4C), RBD with the adjuvant group was 100 times lower than those of the 6HB group (*p* < 0.0004) (Figure 4C), indicating that the candidate vaccine might have the effect of inhibiting in vivo virus replication.

The surface of the lung tissue in group A was covered with a smooth serosa with no obvious abnormality; the lung parenchyma consisted of a large number of alveoli at all levels of the bronchial branches and their terminals in the lungs, and the bronchial structure at all levels had no obvious abnormality. The structure of the group of RBD-6HB is clear; the interstitium, including the connective tissue and blood vessels in the lung, has no obvious abnormality; no obvious pathological changes are found (Figure 4D(a)). Small areas of alveolar wall thickening with a small amount of inflammatory cell infiltration (black arrows) were seen in the lung tissue of the group of RBD; numerous blood vessel congestion (blue arrows) (Figure 4D(b)). The lung tissue of the group of 6HB showed extensive alveolar wall thickening with a small amount of inflammatory cell infiltration (black arrow); a small amount of alveolar expansion (blue arrow) (Figure 4D(c)).

## 3. Discussion

SARS-CoV-2 is a major cause of the global pandemic, and an effective vaccine is urgently required. Subunit vaccines are well-established as a reliable and safe platform effective against various infectious diseases such as hepatitis B, diphtheria, pertussis, shingles, and human papillomavirus [11]. Subunit vaccines contain fragments of pathogenic microorganisms, which are highly purified and immunogenic antigens. Using these purified antigens excludes the risk of post-vaccination infection. An approved subunit vaccine against SARS-CoV-2 would represent an important step in the fight against the COVID-19 Pandemic [12].

Since RBD contains the dominant neutralizing epitopes in the S protein, a vaccine consisting of RBD, rather than full-length S protein, is expected to effectively induce neutralizing antibodies and minimize the production of non-neutralizing antibodies. This study reports on RBD-6HB, a trimerized version of the SARS-CoV-2 spike protein RBD. RBD-6HB could be highly expressed in the baculovirus insect cell expression system. We obtained high-purity RBD-6HB protein with a trimeric structure (Figure 2). It was reported that S-Trimer has robust high-level induction of both humoral and cell-mediated immune responses in rodents and nonhuman primates [13]. We believe that RBD proteins with a trimeric structure may also generate similar immune responses. Meanwhile, it was reported that the SARS-CoV-2 spike antigen as a subunit vaccine for COVID-19 was immunogenic, which can be significantly enhanced by CPG [14,15]. We performed two immunizations using aluminum hydroxide and CPG in combination with RBD-6HB protein and assessed the immunogenicity of the subunit vaccine. According to the results of the specific antibody, it is immunogenic in hACE2 mice. Our research data show that it can effectively induce the body to produce specific antibodies (Figure 3). Furthermore, the group of RBD-6HB proteins with a trimeric structure exhibited better immunogenicity than other groups and could induce higher levels of specific IgG antibodies. The challenge study demonstrated that the RBD-6HB protein group with a trimeric structure could induce an immune response in mice more effectively than the other two groups to protect mice and reduce viral load after the challenge. The viral load in the mice was significantly decreased (Figure 4). Based on the results of the histopathological examination and lung RNA viral load, the protective effect of the candidate vaccine against SARS-CoV-2 was evaluated. RBD-6HB immunized hACE2 mice showed effectual protection, effectively reducing the viral load in the lungs and reducing inflammatory tissue damage. Research indicates that it effectively protects hACE2- mice from the SARS-CoV-2 challenge.

The original severe acute respiratory syndrome coronavirus 2 (SARS-CoV-2) virus that was identified at the end of 2019 had evolved, and various variants emerged. The ones that get more attention include Alpha (B.1.1.7), Beta (B.1.351), Gamma (P.1), Delta (B.1.617.2), and Omicron (B.1.1.529) [16,17,18]. As a result of the continuous emergence of mutants, COVID-19 cannot be completely prevented. However, high vaccination coverage has seen reduced hospitalizations and deaths as a percentage of infection cases, suggesting vaccines may reduce the severity and health impacts of COVID-19 infection [19,20,21,22,23,24,25]. The spread and prevalence of SARS-CoV-2 have seriously affected people’s lives and economic development. Future vaccine studies and development against COVID-19 may learn from the experience of the Recombinant Human Papillomavirus 9-Valent vaccine and develop multivalent subunit vaccines to overcome different variants.

## 4. Materials and Methods

### 4.1. Plasmids, Cells, Virus

The SARS-CoV-2 RBD gene sequence was synthesized from Nanjing GenScript Biotechnology Co., Ltd. and connected to the pFastBac1 vector pFB-SARS-CoV2-RBD-6HB; sf9 cells were preserved in our laboratory; DH10Bac^TM^ competent cells were purchased from ThermoFisher Scientific (Shanghai, China) Co., Ltd. The SARS-CoV-2 strain BetaCoV/Beijing/IME-BJ01/2020 was originally isolated by the State Key Laboratory of Pathogen and Biosecurity.

### 4.2. Preparation of Recombinant Bacmid

The pFB-SARS-CoV2-RBD-6HB plasmid was transformed into DH10Bac competent cells and plated on the medium which had 10 μg/mL tetracycline, 50 μg/mL kanamycin sulfate, 7 μg/mL gentamicin, 100 μg/mL X-gal, and 40 μg/mL IPTG, and the cells were incubated at 37 °C for 48 h. Pick the white colonies to SOC medium, shake them at 37 °C for 3 h and use the universal primers pUC/M13F(CCCAGTCACGACGTTGTAAAACG) and pUC/M13R(AGCGGATAACAATTTCACACAGG) for bacterial liquid PCR identification. In addition, the pFastBac1 empty vector was transformed, screened, and cultured according to the above method and used as a control for recombinant bacmid with an empty vector.

### 4.3. Preparation of Recombinant Baculovirus

When sf9 cells were seeded into six-well plates at a cell density of 2 × 10^6^ cells/mL, the medium was replaced with 1 mL of Grace’s medium per well. Dilute 8 μL of Cellfectin^®^ RII and 3 μg of rB-SARS-CoV2-RBD-6HB in 100 μlof Grace’s medium and mix well. The diluted bacmid and Cellfectin^®^ RII were mixed and allowed to stand at room temperature for 20 min. Add the transfection mixture to a six-well plate, incubate at 27 °C for 5 h, discard the transfection medium and replace it with a complete medium. Incubate at 27 °C until cytopathic changes are observed (cells fell off a lot, and adherent cells were sparse and swollen). The supernatant was recovered, centrifuged at 3000 rpm for 5 min, and filtered with a 0.22 μm membrane to obtain the first-generation (P1) recombinant baculovirus, named rBV-SARS-CoV2-RBD-6HB. 1% volume was inoculated into sf9 cells, and the supernatant was harvested 72 h later, centrifuged, and filtered in the same way as the P1 generation, and the second generation (P2) rBV-SARS-CoV2-RBD-6HB was obtained. rBV-SARS-CoV2-RBD-6HB was inoculated into sf9 cells, and the cells were collected after 72 h of culture. A certain volume of Trizol (Sangon Biotech, China)was added, and total RNA was extracted after being placed at room temperature for 10 min. After reverse transcription, Primers RBD-6HB-F (CCCACCATCGGGCGCGGATCCAACTTAAAAAAAAAAATCAAAATGAAGTT) and RBD-6HB-R (CTAGTACTTCTCGACAAGCTTTTATTCCAGAGTTTCGTT) were used for PCR identification. 

### 4.4. Western Blot

rBV-SARS-CoV2-RBD-6HB was inoculated into sf9 cells, and the cells were collected after 72 h of culture. After adding IP cell lysate (Beyotime, Shanghai, China), the cells were sonicated and centrifuged at 12,000 rpm for 3 min at 4 °C to collect the supernatant. After adding a certain volume of 5 × SDS to the supernatant, SDS-PAGE electrophoresis was performed after 10 min of boiling water bath. After a transfer by semi-dry transfer method, block with 5% skim milk at room temperature for 2 h, RBD and RBD-6HB incubate with anti-RBD mouse polyclonal antibody;6HB incubate with Rb pAb to Strep-tag (1:1000, Abcam, Cambridge, UK, ab76949), then wash with TBST. After that, the membrane was incubated with HRP-labeled Anti-Rabbit IgG (H+L) (1:5000, Shanghai, Beyotime, China) or HRP-labeled Anti-Mouse IgG (H + L) (1:5000, Beyotime, Shanghai, China) for 1 h at room temperature, wash with TBST 3 times, and finally, the band was developed using the GEGEGNOME XRQ enhanced chemiluminescence (ECL) (Thermo Fisher Scientific, Shanghai, China). 

### 4.5. Protein Expression and Purification

The rBV-SARS-CoV2-RBD-6HB recombinant baculovirus was inoculated into the suspension cells, then the cells and medium were collected after 96 h, and the supernatant was collected by centrifugation at 3000 rpm for 5 min at 4 °C. Protein purification was performed using Strep-Tactin^®^ XT Superflow^®^ (IBA) (Figure 5). The purified protein was verified by SDS-PAGE and Western Blot. 

### 4.6. Recombinant RBD-6HB Protein Was Identified

SDS-PAGE loading buffer (non-reducing, 5×) (Cowin Bio., Guangzhou, China) was added to the RBD-6HB protein sample at a ratio of 1:4 and centrifuged at 3000 rpm for 30 s to prepare a non-reduced RBD-6HB protein sample. Meanwhile, SDS-PAGE loading buffer (5×) (Beyotime, China) was added to the protein sample at a ratio of 1:4 after 10 min of boiling water, then centrifuged at 3000 rpm for 30 s to prepare a reduced RBD-6HB protein sample. Finally, we obtained a non-reduced RBD-6HB protein sample and a reduced RBD-6HB protein sample. After centrifugation, an appropriate amount of supernatant was taken with a micropipette and directly added to the 10% SDS-PAGE gel sample wells. After a transfer by semi-dry transfer method, block with 5% skim milk at room temperature for 2 h, incubate with anti-RBD mouse polyclonal antibody, and wash with TBST. After that, the membrane was incubated with HRP-labeled Anti-Rabbit IgG (H + L) (1:5000, Beyotime, China) for 1 h at room temperature, washed with TBST 3 times, and finally, the band was developed using the GEGEGNOME XRQ enhanced chemiluminescence (ECL) (Thermo Fisher Scientific, Waltham, MA, USA). 

### 4.7. Animal Immunization Experiments

Specific pathogen-free (SPF, 6–8 weeks old) Humanized BALB/c mice containing human ACE2 (BALB/c-hACE2) were constructed by Cyagen (Suzhou, China). Mice were housed in SPF stainless steel with a constant atmosphere (22–25 °C, 45–50% relative humidity), natural light cycle, and unlimited feeding and drinking. Mice were monitored three times daily for changes in physical appearance and deaths (if any). Female 6-week-old hACE2 mice (Cyagen, Suzhou, China) were randomly divided into three groups in each experiment and intramuscularly immunized with indicated protein and adjuvant. Each group included 6 BALB/c-hACE2 mice.21d after the first immunity, conduct the booster immunity at the same dose as the first immunity. Grouping and immunization doses are shown in Table 1. The body weight was measured every 2 d after the first immune to monitor the weight change. The blood was collected weekly for specific antibody evaluation (Figure 3A). SARS-CoV-2 specific antibodies were evaluated at the indicated weeks via Enzyme-linked immunosorbent assay (ELISA). The COVID-19 mouse IgG antibody detection kit (Guangzhou Da Rui Biotechnology Co., Ltd. (Guangzhou, China)) was used to evaluate serum samples every week (7, 14, 21, 28, 35 d) after immunization. Serum samples were initially diluted at 1:100, after which 1:2 serial dilutions were prepared, following the supplier’s instructions. 

### 4.8. Immunogenicity Analysis

The COVID-19 mouse IgG antibody detection kit (Guangzhou Da Rui Biotechnology Co., Ltd.) was used to evaluate serum samples every week (7, 14, 21, 28, 35 d) after immunization.

### 4.9. Virus Challenge

All immunized hACE2 transgenic mice were anesthetized with tribromoethanol, and 30 μL of the virus was injected into the trachea at a titer of 1 × 10^5.5^ TCID 50/mL. Studies have shown that intratracheal inoculation is more effective than intranasal inoculation. The virus infection experiment was carried out in the BSL-3 laboratory at Changchun Institute of Veterinary Medicine, Chinese Academy of Agricultural Sciences. After 5 d of challenge, mice were anesthetized by carbon dioxide (CO_2_), euthanized by cervical dislocation, and lung and other organs were collected for further evaluation. Lung tissue was fixed in 4% paraformaldehyde solution to prepare paraffin-embedded sections. Hematoxylin and eosin (H&E) staining was then used to identify pathological changes. The images are then observed and captured through an optical microscope. The remaining lung tissue was ground, then the tissue homogenate was centrifuged, and the supernatant was collected for RNA extraction. Viral RNA from tissue homogenates was extracted by the QIAamp Viral RNA Kit (Qiagen, Hilden, Germany). Viral loads were then detected by the RT-qPCR method using HiScript II One Step qRT-PCR SYBR Green Kit (Vazyme Biotech, Nanjing, China). Primers were designed according to the N gene and E gene of SARS-CoV-2, and the viral loads in the lung tissue were detected by the TaqMan fluorescence quantitative PCR method.

### 4.10. Statistical Analysis

Statistical analysis was performed using GraphPad 8.0 (GraphPad Software, San Diego, CA, USA) with the one-way analysis of variance (ANOVA; two-tailed, confidence intervals (CI) 95%), as indicated by the *p*-value. The results were statistically significant at *p* < 0.05. At least three independent experiments were evaluated for each separate set of assays. The results are expressed as the mean ± standard deviation (SD). 

## 5. Conclusions

In summary, our findings underscore the importance of trimerized SARS-CoV-2 S protein RBD in SARS-CoV-2 Vaccine design and provide a theoretical basis for developing protective vaccines that induce antibodies against the RBD domain.

## 6. Patents

Trimeric receptor binding domain of COVID-19, its preparation method and application (the patent number: 202110184678.X).

## Figures and Tables

**Figure 1 ijms-23-13716-f001:**
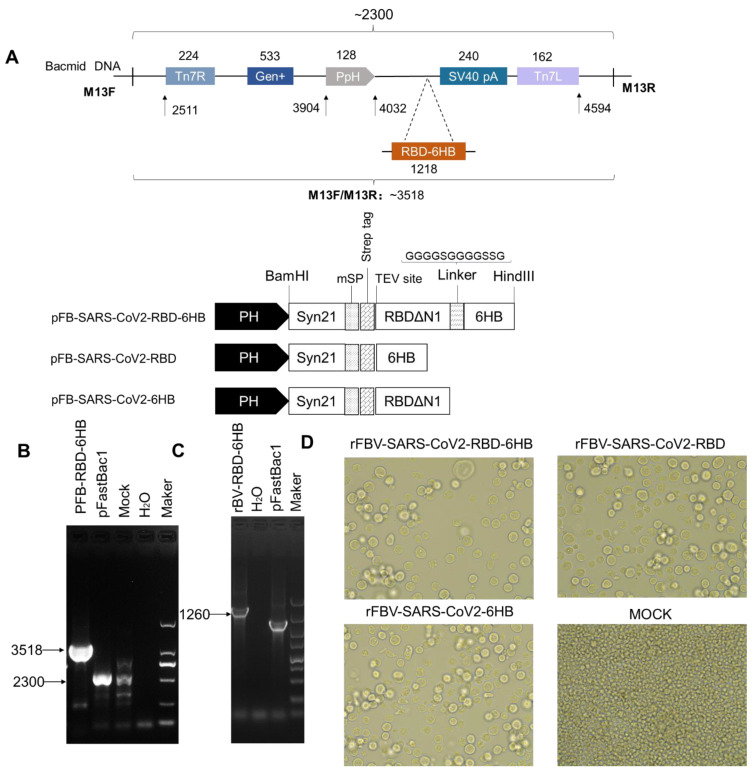
Protein expression and purification. Schematics of the three recombinant bacmid rFB-SARS-CoV-2 (**A**). Identification of recombinant bacmid by RT-PCR (**B**). Identification of recombinant bacmid by RT-PCR (**C**). Microscopic observation at 72 h after SF9 infection with baculovirus (200×) (**D**).

**Figure 2 ijms-23-13716-f002:**
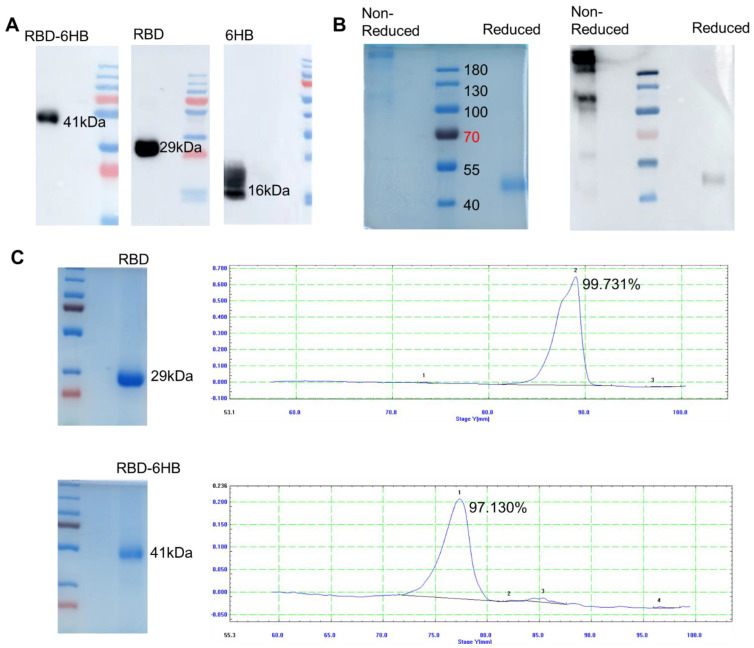
Protein expression and purification. Western blot (**A**). Non-reduced and Reduced SDS–PAGE of RBD- 6HB protein. (left). Non-reduced and reduced Western blotting of RBD- 6HB protein. (right) (**B**). SDS–PAGE and Western-blotting analyses of the eluted RBD and RBD-6HB samples. (The sequence of samples from left to right was Non Reduced RBD-6HB, Non Reduced BXT (negative control), Reduced RBD-6HB, Reduced BXT (negative control) (**C**).

**Figure 3 ijms-23-13716-f003:**
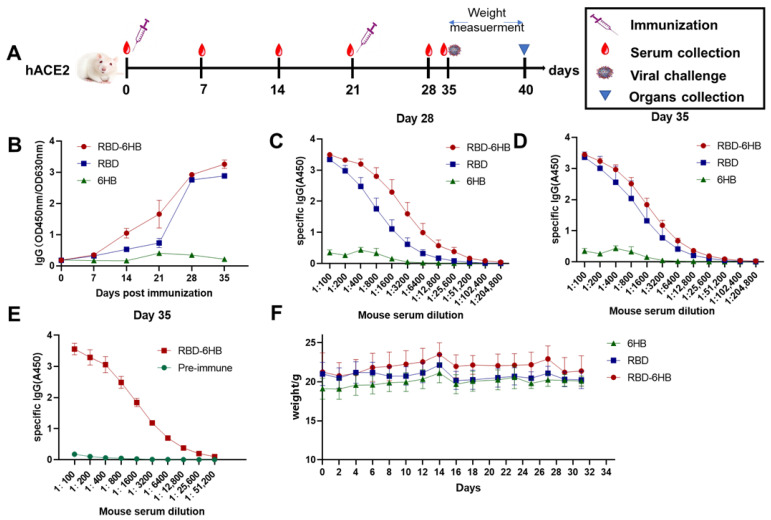
Serum antibody responses after immunization in mice. Blood samples were collected from the tail vein of mice at indicated times after the initial vaccination and used to analyze humoral immune responses using an antibody detection kit. Sera collected from the tail vein of pre-immune mice were used as a negative control. The results are expressed as the mean ± standard deviation (SD) Schematic diagram of mouse immunization (**A**). The mice were immunized with 20 μg recombinant RBD-6HB protein per mouse in the presence of aluminum hydroxide and CPG, compared with the control groups that received the recombinant RBD protein or recombinant 6HB, Sera were collected from the mice 7, 14, 21, 28, and 35 d after the first immunizations and the levels of IgG against the recombinant protein were tested for same serum dilutions using ELISA (Methods) (**B**). Sera were collected from the mice 28 d after the first immunizations, and IgG levels against the recombinant protein were tested for different serum dilutions using ELISA (Methods) (**C**). Sera were collected from the mice 35 d after the first immunizations, and IgG levels against the recombinant protein were tested for different serum dilutions using ELISA (Methods) (**D**). Sera were collected from the mice 35 d after the first immunizations compared with the pre-immune serum. IgG levels against the recombinant protein were tested using ELISA for the same serum dilutions (Methods). (**E**). Measure the body weight before the first immunization and every two d after the first immune to monitor the weight change (**F**).

**Figure 4 ijms-23-13716-f004:**
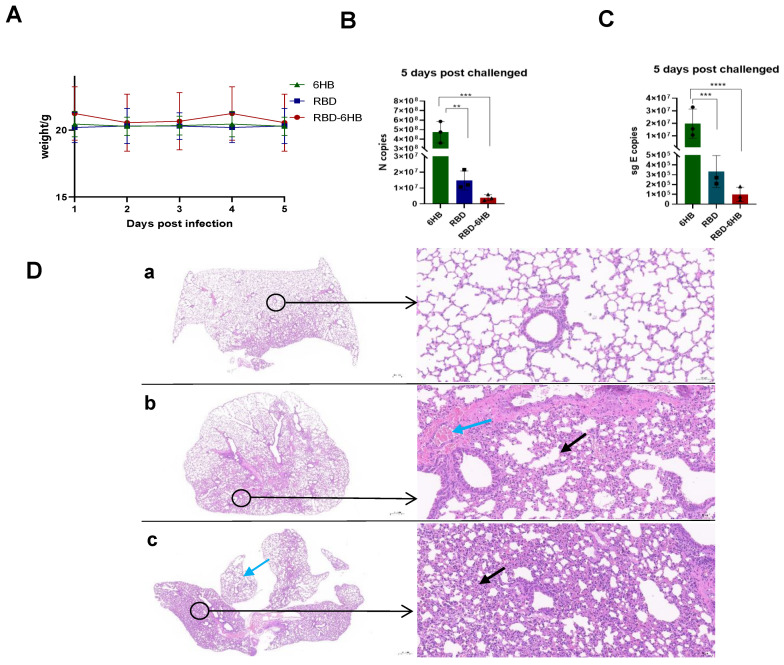
Protection of BALB/c-hACE2 mice by RBD-6HB vaccine immunization. Changes in body weight of mice after challenge (**A**). Viral load analysis of N gene. *p* < 0.01 was summarized with 2 asterisks [* *], *p* < 0.001 was summarized with 3 asterisks [* * *] (**B**). Viral load analysis of E gene. *p* < 0.001 was summarized with 3 asterisks [* * *], *p* < 0.0001 was summarized with 4 asterisks [* * * *] (**C**). Pathological section analysis The left view was 2.0X, and the right view was 20.0X. The group of RBD-6HB (**a**). The group of RBD (**b**). The group of RBD (**c**) (**D**).

**Figure 5 ijms-23-13716-f005:**
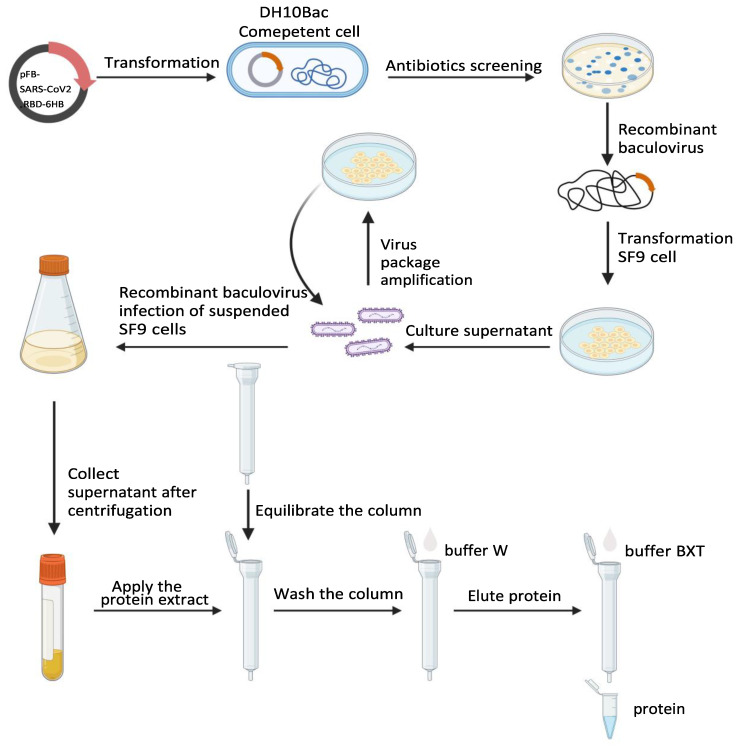
Protein expression flow chart (created by Biorender).

**Table 1 ijms-23-13716-t001:** Immunization groups and doses.

Group	Vaccine	Adjuvant	Animals	Immune Times	Interval Times (Weeks)	Number
A	RBD-6HB	CPG10 μL+ aluminum hydroxide 100 μL + 20 μg	hACE2/B	2	6	6
B	RBD	CPG10 μL+ aluminum hydroxide 100 μL + 20 μg	hACE2	2	6	6
C	6HB	CPG10 μL+ aluminum hydroxide 100 μL + 20 μg	hACE2	2	6	6

Group C was the negative control group. Immunization doses were administered by intramuscular injection in the legs. Serum samples were collected every week after immunization (7, 14, 21, 28, 35 d).

## Data Availability

All data generated or analyzed during this study are included in this published article.

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
