# Peer review of "A Vaccine of SARS-CoV-2 S Protein RBD Induces Protective Immunity"

_ijms, 2022, doi:10.3390/ijms232213716_

Round 1
Reviewer 1 Report
Summary:
The manuscript titled “A vaccine of SARS-CoV-2 S protein RBD induces protective immunity ” describes a novel RBD subunit vaccine that is combined with 6HB and adjuvant. The authors demonstrated vaccine efficacy against COVID disease in the lungs of mice. This new vaccine does elicit a strong IgG antibody response. The manuscript overall is well organized and figures are clear. I have the following comments for your consideration:
Introduction:
The first paragraph is very detailed and well written but it is long. Consider breaking it up to smaller paragraphs. Also it is unclear why the authors are using the 6HB with the RBD. Please elaborate on the advantages to using 6HB in this new COVID vaccine.
Materials and Methods:
Line 186: Intratracheal injection was used for the challenge. Why was this method used instead of intranasal inoculation, which would evaluate vaccine protection of both upper and lower respiratory tract?
Results:
Line 257-258: The authors conclude that no body weight change was observed after immunization indicating the vaccine was not irritating. It is unclear how irritation is tied to changes in body weight. Please elaborate.
Author Response
Dear Reviewer,
We would like to express our appreciation to you for the helpful suggesting in improving our manuscript. We have carefully considered the comments and revised the manuscript according to the comments. All revisions to the manuscript were marked up using the MS “Track Changes” function and the version with track changes were resubmitted. The attachment is a point-by-point response to all those comments and a list of changes we have made to the manuscript.
And the revised version has been polished by native speaker.
Thank you again for your special efforts.
With best regards,
Yours sincerely,
Prof. Chang Li Ph.D.
Changchun Institute of Veterinary Medicine, Chinese Academy of Agricultural Sciences, Changchun, Jilin Province, China;
E-mail: [email protected]

Reviewer 2 Report
In the paper named “A vaccine of SARS-CoV-2 S protein RBD induces protective immunity” author make a study in order to improve the actually methods to make covid-19 vaccines. As the receptor binding domain (RBD), located at the spike (S) gene, is responsible for binding to the angiotensin converting enzyme 2 (ACE2) receptor of host cell, this can be a good antigen candidate in vaccines creation. In this work, author study if the six-helix bundle (6HB) ‘molecular clamp’ that is a novel thermally-stable trimerization domain, derived from a segment of human immune deficiency virus (HIV) gp41 protein is even better candidate in vaccines creation. The result of this study shown that the RBD-6HB protein combined with AL/CpG adjuvant can stimulate animals to produce sustained high-level antibodies and establish an effective protective barrier to protect mice from challenges. Therefore thisfindings highlight the importance of trimerized SARS-CoV-2 S protein RBD in the design of SARS-CoV-2 vaccines and provide a rationale for the development of a protective vaccine through the induction of antibodies against the RBD domain.
Only minor points are required
1) In material and method sections and in results author say that cytostatic changes are observed in the cells, please can give more details about this?
2) In Point 2.4 authors say that the cells extraction is made adding IP lysate before sonication, what is the IP lysate composition?
3) In point 2.5, how authors perform the plasmid inoculation in the suspension cells?
4) Point 2.6 is very confusing, please clarify. Information about the microinjection, the SDS page % and other important data are missing
5) In Point 2.7 the number of mice used in each treatment are missing
6) Paragraph before line 166 and 167 is confusing
7) In line 174 author say that the serum sample were obtained every week (7,14,21,28,35,42d ) however in figure 2 A the time course is little different. The same happens in table 1 where author state that the time course of serum collection is (7,14,21,28,35d).
8) Only lung tissue was analyzed? Have author make some analysis in other tissue?
9) In point 3.1 authors say that “After inoculation of rBV-SARS-CoV2-RBD-6HB into sf9 cells, it was observed obviously cytopathic effect (CPE) (cells fell off a lot, and adherent cells were sparse and swollen) under microscope at 72 h (Fig. 2D).” but in figure 2D it is seems that the cytostatic are in all the treatment, only MOCK is without effect. Please can explain this issue?
10) In point 3.9 author treat the cell extract with reducing and non reducing buffer and shown the gel images in figure 3.B however the difference in the distaining in the gels make difficult to compare RBD and RBD-6HB. In both cases a big band can be seen in non reduced part.
11) In line 257 author say that the dilution 1:25600 shown strong antibody response in mice, but in the figure 4 this is difficult to see. Please can explain this point.
Author Response

(The authors gave the same response as above.)
